# Analysis of Heat Shock Proteins Based on Amino Acids for the Tomato Genome

**DOI:** 10.3390/genes13112014

**Published:** 2022-11-02

**Authors:** Meshal M. Almutairi, Hany M. Almotairy

**Affiliations:** National Center of Agricultural Technology, Sustainability and Environment, King Abdulaziz City for Science and Technology KACST, Box 6086, Riyadh 11442, Saudi Arabia

**Keywords:** amino acids, codon adaptation index, relative synonymous codon usage, modified relative codon bias, principal component analysis

## Abstract

This research aimed to investigate heat shock proteins in the tomato genome through the analysis of amino acids. The highest length among sequences was found in seq19 with 3534 base pairs. This seq19 was reported and contained a family of proteins known as *HsfA* that have a domain of transcriptional activation for tolerance to heat and other abiotic stresses. The values of the codon adaptation index (CAI) ranged from 0.80 in Seq19 to 0.65 in Seq10, based on the mRNA of heat shock proteins for tomatoes. Asparagine (AAT, AAC), aspartic acid (GAT, GAC), phenylalanine (TTT, TTC), and tyrosine (TAT, TAC) have relative synonymous codon usage (RSCU) values bigger than 0.5. In modified relative codon bias (MRCBS), the high gene expressions of the amino acids under heat stress were histidine, tryptophan, asparagine, aspartic acid, lysine, phenylalanine, isoleucine, cysteine, and threonine. RSCU values that were less than 0.5 were considered rare codons that affected the rate of translation, and thus selection could be effective by reducing the frequency of expressed genes under heat stress. The normal distribution of RSCU shows about 68% of the values drawn from the standard normal distribution were within 0.22 and −0.22 standard deviations that tend to cluster around the mean. The most critical component based on principal component analysis (PCA) was the RSCU. These findings would help plant breeders in the development of growth habits for tomatoes during breeding programs.

## 1. Introduction

Temperature is the primary environmental variable playing a significant role in plant growth and survival. Increasing the temperature to about 12 °C above the threshold point reduces plant productivity by affecting embryogenesis, germination, and fruit development. Tomato is sensitive to heat stress; increasing a few degrees above the average day temperature (25 °C) can impair reproductive organs, resulting in the failure of fruit setting. Heat stress reduced tomato production in both open field and non-controlled growing conditions. The limitations in understanding the genetic basis of heat tolerance traits limits the breeding of tomatoes, which requires a depth of knowledge about their genetic architecture [1]. Tomatoes respond to heat stress using a group of polypeptides during cellular activity known as heat stress transcription factors (HSFs). The factors that are molecular chaperones mediate the rapid accumulation of heat shock proteins during biotic and abiotic stresses, which help plants to protect themselves from damage and degradation of proteins and maintain optimal growth [2]. A small gene family code called Hsfs exists as inactive proteins with up to five members in plants. Hsfs contain two clusters of basic amino acid residues considered as K/R motifs for localization. Moreover, these factors are classified into different families localized to cellular compartments such as the nucleus, endoplasmic reticulum mitochondria, and chloroplasts. These HSFs prevent aggregation and translocation by binding to denatured proteins [2,3]. Stress causes activation with oligomerization when HsfA1 binds with its target sequences, where present, in the promoter of the genes. Some studies showed that the heat shock proteins’ chloroplast is involved in plant thermotolerance, which protects the photosystem II. A few of the chaperones such as Hsp17–CII, Hsp70, and Hsp90 are involved in the second part of the Hsf cycle that leads to the restoration of the inactive state. More than 365-trained deep learning models of all tomatoes’ genomes were applied for better predictions [4,5,6,7]. 

Tomatoes that belong to the family Solanaceae have a unique flavor and high nutritional value. Tomatoes are diploid, with a genome size of about 950 Mb and 12 chromosomes. During the domestication and breeding processes, tomatoes dramatically change fruit shape, color, and plant tolerance according to a biotic and/or an abiotic stress [8]. More than 1000 tomato varieties and wild species have been sequenced [9]. Tomatoes are extremely sensitive to heat stress, which causes abortion of the male gametophyte, reflecting a reduction in fruit set and yield when there is an increase in daytime temperature above 26 °C. High temperatures (44 °C) cause cell death due to released cytochrome and induced caspase-like enzymes [10]. This extremely high temperature affects membrane fluidity and metabolic mechanisms, which cause the overproduction of reactive oxygen species and oxidative stress [11,12]. Heat shock proteins are produced in cells due to exposure to stress. It was documented that these proteins protect cells from destructive effects during the synthesis of proteins. Some studies have evaluated heat tolerance in tomatoes using different parameters such as the phenotypic index, while others have suggested the investigation of codon usage patterns. Thus, codon usage patterns provide one insight into the impact of high temperatures on tomato proteins. The arrangement of genetic code responds to the synthesis of all protein molecules, which are revealed by using sixty-one codons to encode 20 amino acids. These codon usage patterns vary among genes. However, it is still challenging to decide the most common dominant codon bias due to compositional constraint and codon–anticodon interaction [12]. 

Amino acids are a fundamental measurement for the plant growth stage due to their connection to protein structures, functions, and much metabolic flux [13]. Thus, these amino acids are essential for establishing the basics of genetic information. In addition, a genetic code uses 61 codons, corresponding to 20 amino acids encoded to more than one codon, that can be revealed as a synonymous codon except for both methionine and tryptophan, which are encoded by only one codon [14,15]. Bias in synonymous codons may arise because of natural selection or mutation [16,17]. It is well known that genetic codes direct all proteins in the DNA genome. However, codon usage pattern varies among organisms and genes. Thus, this variation suggests different factors that influence the codon usage pattern. Several reports explained the codon distribution in the proteins among different synonymous codons. These factors are nucleotide composition, tRNA abundance, the structure of the protein, and translation processes. To explain more about the codon usage pattern, the codon adaption index (CAI) is one of the best methods for calculating the frequency of the synonymous codon for more understanding of gene expression, new gene discovery, molecular mechanism, and genetic engineering. It was found that heat shock proteins were accumulated in the mRNA. However, there is a lack of understanding of the genetic processes involved in heat stress during tomato growth. Therefore, this research aimed to understand the mechanisms of heat shock proteins in tomatoes by investigating heat shock proteins through analysis of amino acids [18,19,20,21,22].

## 2. Materials and Methods

The heat shock protein sequences for the *Solanum lycopersicum* cultivar Heinz 1706 were retrieved from the National Center for Biotechnology NCBI (https://www.ncbi.nlm.nih.gov/nuccore/?term= (accessed on 7 June 2021)). These 19 proteins, with their accession numbers, were provided (Table 1) [23,24,25,26,27,28]. 

These proteins were derived from a whole-genome shotgun sequence supported by EST evidence. In short, tomato genomes are an international consortium and have been extensively characterized to better understand genetic diversity [29,30]. All proteins were checked based on the following considerations, such as the presence of START and STOP codons, and removed duplicated sequences [18]. 

A variety of analysis tools were utilized, such as the codon adaptation index (CAI) given by Sharp and Li [31], and RSCU provided by Yu et al. [22]. In addition, guanine-cytosine content (GC) content, relative codon bias strength (RCBS), and MRCBS were applied in this study [32]. 

The following formula represents the codon adaptation index: CAI = (∏1^*N* wi) 1/*n*, where *N* is the number of a codon in the gene and relativeness wi is described as wi = fi/f (aa, max). Fi is the frequency of the ith codon, and f (aa, max) is the maximum frequency of the codon most often used to encode amino acid aa [31]. 

Relative synonymous codon usage can be estimated using the following formula: RSCU = Xij/(∑j^niXij), where xij shows the frequency of codon j coding for the ith amino acid and ni represents the number of the synonymous codon encoding the ith amino acids. The interpretation for the RSCU is that an RSCU value greater than 0.5 has more frequency, and an RSCU value smaller than 0.5 indicates less frequency in a gene for a particular codon. Whereas RSCU > 1 indicates an overrepresented codon for the corresponding amino acid [33]. 

Modified relative codon bias measures the expression level of a gene [32]. MRCSB shows in the formula MRCBS =∏ (i=1) *N*(MRCBS_xyz_). And (MRCBS_xyz_) = (RCBS_xyz_)/(RCBS aa, max). And (RCBS_xyz_) = fxyz/(f_1(x)_ f_2(y)_ f_3(z)_), fxyz is the normalized codon frequency of a codon xyz and f*n*_(m)_ is the normalized frequency of base m at codon position n in a gene. The (RCB aa, max) reflects the maximum value of the RCBS of a codon encoding the same amino acid aa. The score of MRCBS ranges from zero to one. 

The data from RSCU, GC content, RCBS, and MRCBS were computed integrally using Microsoft Excel. Two-way analysis of variance was computed using the PROC GLM procedure at a significate level of the *p*-value (0.001) in SAS computer packages version 9.2 for Windows (SAS Institute Inc., Cary, NC, USA). In addition, analysis of principal components helps to re-set data into a few data sets computed using PROC PRINCOMP in SAS to analyze total variations.

## 3. Results

The ANOVA test for the MRCBS data reflects the significant difference among 21 amino acids, 64 codons, and 19 sequences with a *p*-value < 0.0001 (Table 2). For example, the maximum means ±SD across all sequences in MRCBS was 0.67 ± 0.36 for seq14. In contrast, the maximum value for RSCU was 0.28 ± 0.22 in both sequences 1 and 19. In addition, the maximum means ±SD for the amino acids asparagine, tyrosine, and tryptophan were 0.87 ± 0.17, 0.85 ± 0.24, and 0.80 ± 0.44, respectively. Thus, it was found that all parameters for testing a normal distribution were significant, following a uniform distribution.

### 3.1. Genome Sequences Based on the Codon Adaptation Index and GC%

The highest length among sequences was found in seq13 and seq19, with 3982 and 3534 base pairs, respectively (Table 2). The CAI ranges from zero to one, and the higher values of CAI were found in seq19 (0.80) while the lower ones were found in seq10 (0.65). It was reported that sequence19 contained a family of proteins known as HsfA that have the domain of transcriptional activation for tolerance to heat and other abiotic stress [24]. Under stress, hetero-oligomers were formed by the interaction between HsfA1 and HsfB1. It was found that HsfA1 plays an important role in protecting cellular homeostasis under heat stress. This illustrates that the HsfA1 group is a master regulator of heat-stress response in tomatoes, as reported by Mishra et al. [34]. In addition, the HsfB1 expressed significantly with increased heat stress, as known by Kumar et al. [35]. These heat stress transcription factors (Hsfs) are the terminal components for the activation of genes responsive to heat and other stresses. The HsfA1, HsfA2, and HsfB1 are located between the L1 and L2 positions and on C4 at the DNA-binding domain in the tomato genome. However, the HsfA1 is different from all others by an insertion of 21 amino acid residues in the linker. Moreover, protein resulted in higher GC% in seq2 (44.15), and the length of sequences demonstrates the impact of amino acid composition on synonymous codons. This impact is essential to illustrate the limitation of the expected value of CAI based on the sequence. Contrary to previous studies, the length of sequences did not affect either GC% or CAI. The lowest GC% content was 33.29% in Seq12, while the highest was 44.15% in Seq2. These proteins ranged from the most increased length, about 3982 base pairs in Seq13, to the smallest length (819 base pairs) in Seq6. A few reports illustrated that the GC content of rRNA and tRNA showed a strong positive correlation with their optimal growth temperature. 

The GC content is an important qualitative aspect in the genomic nucleotide reflected genomic architecture. The importance of this GC content is in conferring stability to higher order structure of DNA and RNA transcripts. Here, it was observed that GC% for heat shock protein (HsfA1) was 37.72%, reflecting the moderated stability of DNA compared to the AU base pairs that were less stable. The second higher of GC% was Seq14, with accession no XM_015230190.2, known also as heat shock protein (Hsf8). The variation in GC% within the identical genomes contributed to mapping genes and biased recombination associated with DNA repair. It was reported that some repeated sequences are indeed GC rich, include the tandemly repeated and interspersed rDNA. Sometimes, the repairing of mismatched base pairs is hypothesized to be biased to certain nucleotides during DNA repair or gene transcription [36]. Carels and Bernardi [37] reported that proteins produced by GC-rich and GC-poor genes showed functional differences. Both of these GCs were different in amino acid composition. In addition, it was found in grasses that fewer introns control GC-rich genes. GC content and genome size showed a quadratic relationship due to the biochemical costs of GC base synthesis [38].

### 3.2. The Codons of Amino Acids Based on RSCU Analysis

The synonymous codon usage was analyzed using the above, given by Yu et al. [22]. RSCU is calculated as the observed frequency of the codon divided by its expected frequency. For more illustration, an RSCU value bigger than 0.5 indicates a more frequently used codon. While an RSCU value less than 0.5 indicates a less frequently used codon (Figure 1). It can be seen that histidine, asparagine, aspartic acid, cysteine, phenylalanine, and tyrosine have RSCU values bigger than 0.5. Asparagine (AAT, AAC), Aspartic acid (GAT, GAC), Phenylalanine (TTT, TTC), and Tyrosine (TAT, TAC) have second positions as A and T, while the third position in these amino acids has either T or C. This result reveals that the high RSCU values were rich with these two codons, A (adenine) and T (thymine). These second nucleotide positions should emphasize the nature of the encoded amino acids that determine the transmission of information from DNA to mRNA to protein.

Moreover, RSCU values less than 0.5 were considered rare codons such as lysine, glycine, threonine, isoleucine, proline, strop codon, valine, leucine, glutamic acid, arginine, serine, and glutamine, which affect the translation [28]. Thus, selection could be effective by reducing the frequency of expressed genes under heat stress. The normal distribution of RSCU is important to reflect the population distribution (Figure 2). This distribution is known as symmetric around the means. The means and standard deviation of the RSCU values were 0.27 and 0.22, respectively. Therefore, 68% of values drawn from the standard normal distribution are within 0.22 and −0.22 standard deviations, which tends to cluster around the mean. 

### 3.3. The MRCBS Analysis of Codons for Heat Shock Proteins

The MRCBS values were calculated based on all tomato proteins under heat shock to obtain the most gene expression for several amino acids that varied from one to 0.14 (Figure 3). The threshold of the MRCB values was 0.62 greater than this threshold and was taken as the standard for determination of the high gene expression. Therefore, the increased gene expression of the amino acids under heat stress was histidine, tryptophan, asparagine, aspartic acid, lysine, phenylalanine, isoleucine, cysteine, threonine, valine, glutamic acid, and glycine. Other amino acids that less than the threshold (0.62) were considered as less gene expression. Some significant scores were found in some codons that contribute to gene expressions such as (CAT) As (Ala), TGC (Arg), ACA (Arg), TAC (Asp), AGA (Cys), GGC (Glu), CTG (Gly), CAA (Ile), TAA (Ile), GGG (Leu), CCG (Lys), GAC (Pro), GTC (Ser), CAG (Ser), CGA, CGT (Stop), TAG, GCG (Thr), GGA (Tyr), and CTA (Val) < 0.05 (the Z test). These significant codons may affect translational efficiency [39,40]. Most of the high gene expressions were unknown to their characterized hypothetical genes. Thus, genes with high-predicted expressions may have important functions that were unknown. These proteins with their codons may provide targets for identifying critical features of tomato genomes under heat stress [41]. 

### 3.4. Principal Component Analysis

Principal component analysis is useful for identifying unknown trends in tomato genomes and simplifying the description of the tomato genome by analyzing the structure of the observations and variables. The two dominations in Figure 4 show the total variability for 19 mRNA sequences related to heat shock proteins. The first domination accounts for 80.5% of the total variance, while the second component accounts for 6.2% of the total variance. The eigenvalue greater than 1 was the RSCU with a value of 1.94, which correlated to PC1 with 64.7% of the total variance. While the eigenvalues that were less than one were in both MRCBS with 0.80 and RCBS with 0.26, which together account for 35.2% of the total variance that correlated to PC2 and PC3, respectively. In addition, the most important component of the tomato genome was the RSCU and MRCBS. Thus, the most variation was explained in these first and second components. While the RCBS had less importance in revealing the variation and its eigenvalue was 0.26. The higher contribution of variations among 19 mRNA sequences in the tomato genomes are from the following sequences: Seq14, Seq2, Seq3, Seq16, and Seq15, while the lower contributions of variations are from Seq18, Seq4, Seq19, Seq8, and Seq6. Other sequences were between the higher and lower contributions to total variation.

## 4. Discussion

A sustainable crop system such as tomato is important for its food and economic value. Global temperature shows an increasing trend, making heat stress a critical issue; meanwhile genetic and physiological responses to this stress should be considered. The tomato is known to be extremely sensitive to heat stress, reflecting a reduction in fruit set and yield when there is an increase in daytime temperature above 26 °C [11]. The tomato responds to a high temperature by inducing the synthesis of a group of polypeptides known as heat shock proteins ranging from about 15 to 42 kD. Out of 20 amino acids, 18 are encoded with more than two synonymous codons at the level of the genetic code. Some codons are translated more efficiently and accurately due to these codons being favored over others by natural selection [15]. 

Predicting gene expression from nucleotide sequences is considered fundamental knowledge for modern bioinformatics. This gene could be turned on for desired genes or edited in genes for more specificity. This gene expression is controlled by several factors, such as protein biosynthesis, mutation, and natural selection. These nucleotides and gene expressions would increase the translation efficiency of the gene population. Due to several factors, the CAI values may indicate some nuclear singles in codon bias. One example of these factors is the neutral mutation press for shaping these codons’ usage, while another example is the translation levels [4,5,6]. 

The GC content helps breeders to predict conserved amino acid patterns [1]. Several studies have argued that GC content may reflect the primary influence on codon usage patterns that change the expression profile. The GC content was reported as one of the principal influences of codon usage patterns that changed the gene expression. Thirty optimal codons ended with A, T, or G with 13, 15, or 2, respectively. It was stated that the GC index is more sensitive to rare amino acids, and thus it is poorly informed and could not provide details for codon usage; however, a small gene that is less than 100 codons could be used in the GC index more informatively [16]. 

## 5. Conclusions

This paper showed that RSCU, RCBS, and MRCBS might be useful tools for predicting the amino acid frequency and gene expression during the growth stages of tomatoes. The idea of supporting our method and finding is based primarily on the responsible codon usage pattern for the regulation of gene translations that occur during protein translation. This objective would help plant breeders in the development of growth habits for tomatoes. It also helps to demonstrate significant heterogeneity in codon usage among genes in tomato genomes [26,27].

The CAI and GC% were used to measure the synonymous codon usage bias for the mRNA of heat shock proteins in tomatoes. The values of CAI ranged from 0.80 in Seq19 to 0.65 in Seq10. Protein resulted in higher GC% in Seq2 (44.15), and the length demonstrates the impact of amino acid composition on synonymous codons. An RSCU value bigger than 0.5 indicated a more frequently used codon, while an RSCU value less than 0.5 indicated a less frequently used codon. RSCU values bigger than 0.5 were observed in histidine, asparagine, aspartic acid, cysteine, phenylalanine, and tyrosine have asparagine (AAT, AAC), aspartic acid (GAT, GAC), phenylalanine (TTT, TTC), and tyrosine (TAT, TAC). In addition, this result reveals that the high RSCU values were rich with these two codons, As (adenine) and Ts (thymine). The overall results here indicate that the highest genes expressed of the amino acids were histidine, asparagine, aspartic acid, cysteine, phenylalanine, tyrosine, and asparagine. In addition, the result of the codon adaption index can help for more confidence in predictions. More studies are required to understand the amino acid network metabolism, such as biochemical, molecular, and genomics [34,42]. 

Gene function can be determined through natural selection and could be developed for adaptability to the environment. It was found that arginine and proline metabolism, alanine, aspartate, and glutamate were significantly disturbed by drought stress [43,44,45].

Our results provide the first understanding of the tomato genome to facilitate the study of genetic mechanisms in tomato biology. 

## Figures and Tables

**Figure 1 genes-13-02014-f001:**
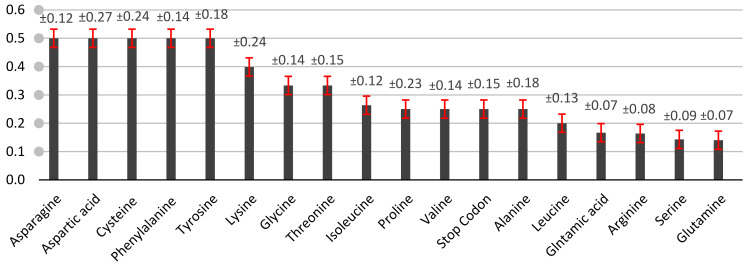
The RSCU mean in heat shock protein sequences based on mRNA for the *Solanum lycopersicum* cultivar Heinz 1706, excluding Methionine and Tryptophan, with standard error shown by the red line and label with standard deviation (±).

**Figure 2 genes-13-02014-f002:**
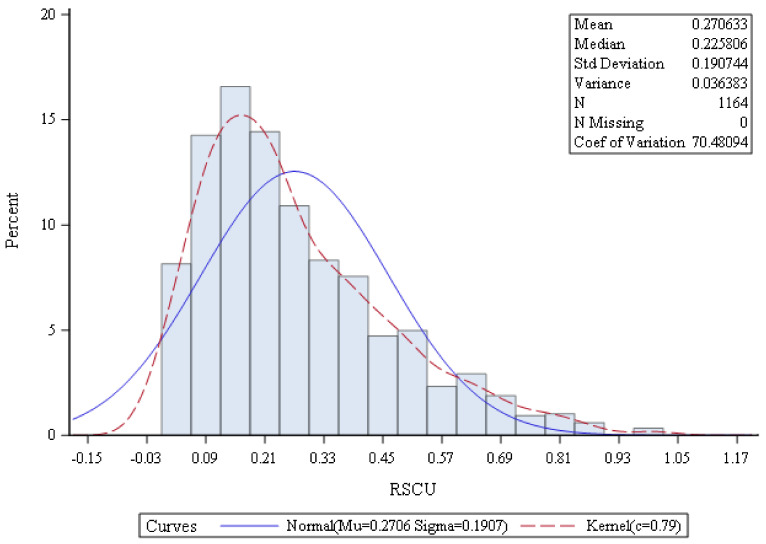
The normal distribution of the RSCU values in heat shock protein sequences based on mRNA for the *Solanum lycopersicum* cultivar Heinz 1706.

**Figure 3 genes-13-02014-f003:**
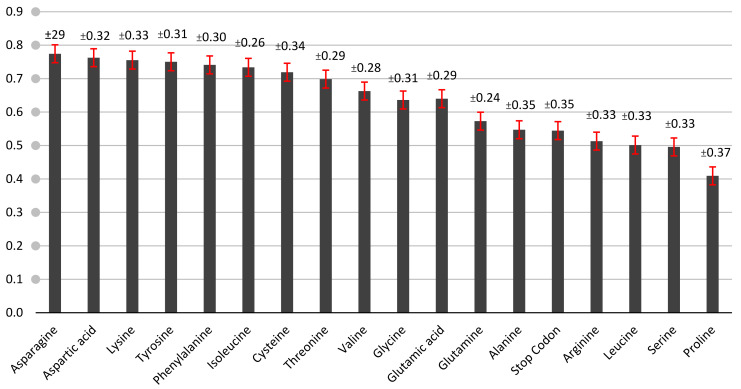
The MRCBS mean in heat shock protein sequences for the *Solanum lycopersicum* cultivar Heinz 1706, excluding Methionine and Tryptophan, with standard error shown by the red line and label with standard deviation (±).

**Figure 4 genes-13-02014-f004:**
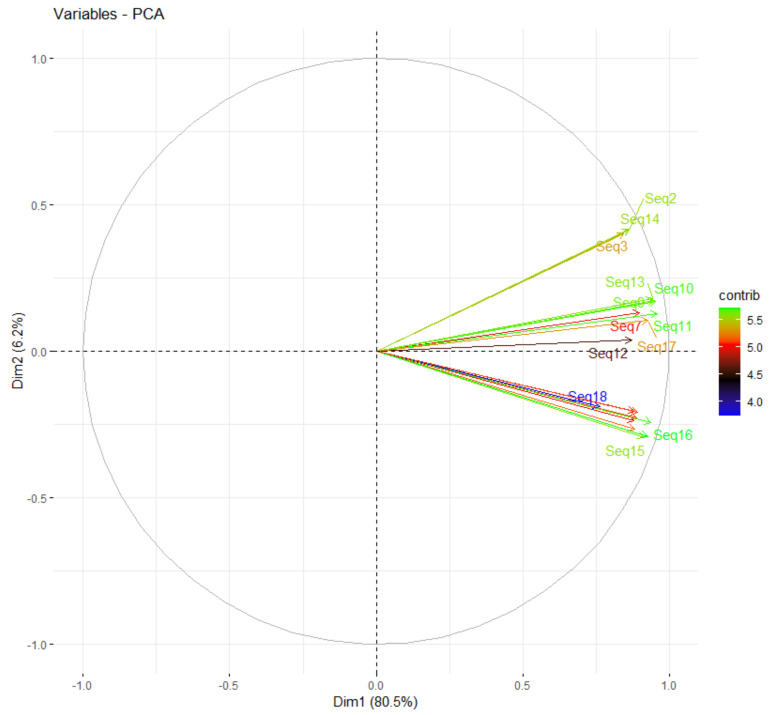
The percentage of two dominations explains variability through principal components for 19 mRNA sequences related to heat shock proteins for the *Solanum lycopersicum* cultivar Heinz 1706, the right legend shows the contributions of variables to PCs. Dim1 expresses dimension 1 while Dim2 expresses dimension 2.

**Table 1 genes-13-02014-t001:** The basic of *Solanum lycopersicum* sequence attributes under heat shock.

Sequences	Accession No.	CAI	GC%	Length (bp)
1	XM_010327263.3	0.77	41.99	2023
2	NM_001309248.1	0.66	44.15	1973
3	NM_001321563.1	0.68	41.71	2539
4	NM_001246851.2	0.78	42.14	2589
5	NM_001247296.2	0.77	40.39	822
6	NM_001320262.1	0.76	39.32	819
7	NM_001321450.1	0.75	39.28	1647
8	XM_026032687.1	0.65	38.75	2080
9	XM_010326728.3	0.76	38.83	2127
10	XM_026032686.1	0.65	39.08	2165
11	XM_026032685.1	0.76	37.92	2379
12	NM_001247047.2	0.78	33.29	1673
13	NM_001247134.1	0.66	37.72	3982
14	XM_015230190.2	0.70	44.01	1977
15	XM_027919128.1	0.68	40.06	1343
16	XM_027919127.1	0.67	38.49	1642
17	XM_015208371.2	0.69	40.13	1687
18	XM_027913778.1	0.67	43.82	1376
19	NM_001247342.2	0.80	40.95	3534

CAI stands for the codon adaptation index, GC% stands for guanine-cytosine content, and bp is base pair.

**Table 2 genes-13-02014-t002:** The ANOVA for MRCBS, amino acids and codons for the *Solanum lycopersicum* cultivar Heinz 1706.

Source	DF	Anova SS	Mean Square	F Value	Pr > F
Sequences	18	1.69	0.09	3.39	<0.0001
Amino acids	20	23.09	1.15	41.58	<0.0001
Codons	63	82.47	1.31	47.14	<0.0001

## Data Availability

Not applicable.

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
