# Peer review of "Analysis of Heat Shock Proteins Based on Amino Acids for the Tomato Genome"

_genes, 2022, doi:10.3390/genes13112014_

Round 1

Reviewer 1 Report

Some minor corrections are shown below:

 The scientific name on Line 88, 154 and 156 should be corrected. (Solanum lycopersicum)

The literature display on Line 168 should be corrected. (Brown and Jiricny 1988)

Explanations of abbreviations should be written under the tables.

In principal component analysis,

the eigen values of the obtained sub-principal components (PC1, PC2, ...PCn);

which of these have an eigenvalue greater than "1";

and which characters |0,3| is greater than the eigen value; 

must be reported in the article with a few sentences.

Author Response

  1. In the line 88 the scientific name of tomato has been changed and corrected "Solanum lycopersicum".
  2. in the line 168 the reference has been removed "(Brown and Jiricny 1988)".
  3. the abbreviations has explained under tables.  
  4. I explained the eigenvalue and total variance in principle components based on PROC PRINCOMP which is equivalent to PROC FACTOR. "The eigenvalue that greater than 1 was RSCU with value 1.94 that correlated to PC1 with 64.7% of total variance. While the eigenvalues that less than one were in both MRCBS with 0.80 and in RCBS with 0.26 which together account for 35.2% of the total variance that correlated to PC2 and PC3 respectively."

Reviewer 2 Report

The Introduction, results and discussion must be improved of the manuscript entitled "Analysis of Heat Shock Proteins Based on Amino Acids for Tomato Genome".

- More number of analysis must be carried out in results part.

- The scientific name must be written in a proper way. 

Reviewer 3 Report

Dear authors,

I carefully checked the manuscript content, and it is suitable for publication after the following amendments. Please consider all comments and provide a point-by-point answer to all queries. Ignoring transparent answers to the requested comments may decrease the chance of your paper reaching acceptance. In the current style, the paper data and results required further comparisons with reference model genome data to reach a point where the reviewer can decide on the final destiny of this paper.

General comments:

1-   Please add some specific details to the introduction section in line with the research you conducted there. Please also cite the previous works published on the discussed topic.

2-   Please provide a flowchart for the M&M section.

3-   Please use another graphical representation to show figure 1 details. The data represented here can be statistically expressed to provide essential information for academic readers.

4-    Please subject the previous comment also to figure 3.

5-   Please improve figure 4 and other figures captions. Critical explanations should be added to each figure to interpret the data in detail.

Specific comments.

1-   The authors only recruited 19 protein sequences or genomes? If the authors used protein sequences tools specifically developed for proteome analyses have not been applied here. Please carefully revise the M&M section and use other tools for the evaluation of selected protein sequences.

2-   Which reference genome(s) were used for comparing the data?

3-   To enhance the reliability of the given data here, please also use the Arabidopsis reference genome to compare the produced statistical data for studied sequences. In such studies, homologs and orthologous sequences from reference model genomes should also be taken into consideration to enhance the quality of reported data.

4-   Please check the sequence length of table 1. For example, the authors assigned 2022 bp length for XM_010327263.3 while it is 2023. Please provide all details correctly. For other records, the data should be rechecked.

5-   Please construct a phylogenetic tree for the evaluated sequences to know whether there are differences between the studied proteins or not. Please use a sister group and an outgroup while you try to make a phylogenetic tree for the studied sequences.

6-   Please include the homologous and orthologous statistics from reference model genomes to re-analyze the data reported in figure 4.

7-   Please delete the similarities between the introduction and discussion text and use updated references and critical explanations to describe the novelty and benefits of this study.

8-   Would you please kindly determine how your manuscript output can be applied for practical projects on tomato genomes? This can be considered by adding some lines to the end of the discussion or concluding remarks.

9-   Please re-write the concluding remarks and transfer the cited references to the discussion section. Please note that concluding remarks should be summarized and only represent the novel finding of this study. 

Round 2

Reviewer 3 Report

I have no further comments on this world. Wishing the respected authors all the bestt.